# Healthy Eating Index-2015 Is Associated with Grip Strength among the US Adult Population

**DOI:** 10.3390/nu13103358

**Published:** 2021-09-25

**Authors:** Galya Bigman, Alice S. Ryan

**Affiliations:** 1The Baltimore Geriatric Research, Education and Clinical Center, Veterans Affairs Maryland Health Care System, Baltimore, MD 21201, USA; 2The Baltimore Veterans Affairs Medical Center, Division of Gerontology and Palliative Medicine, Department of Medicine, University of Maryland School of Medicine, Baltimore, MD 21201, USA

**Keywords:** sarcopenia, grip strength, muscle strength, healthy eating index, protein, dietary quality, eating pattern

## Abstract

The Healthy Eating Index-2015 (HEI-2015) was designed to reflect adherence to the 2015–2020 Dietary Guidelines for Americans (DGA). The study aims to examine the association between HEI-2015 and grip strength in a nationally representative sample of the U.S. adult population. This cross-sectional study used data from the National Health and Nutrition Examination Surveys of 2011–2014. Low grip strength was defined as <35.5 kg for men and <20 kg for women. HEI-2015 was computed from two days of 24-h dietary recalls and comprised 13 components. Each component was scored on the density out of 1000 calories and summed to a total score divided into quartiles. Weighted logistic regressions examined the study aim while controlling for associated covariates. The sample included 9006 eligible participants, of those, 14.4% (aged 20+ years), and 24.8% (aged ≥50 years) had low grip strength. Mean (±SD) HEI-2015 total score was 54.2 ± 13.6 and in the lowest and highest quartiles 37.3 ± 5.1 and 72.0 ± 6.5, respectively. In the multivariable model, participants in the highest vs. lowest HEI-2015 quartile had 24% lower odds of having low grip strength (Odds Ratio (OR) = 0.76; 95% CI: 0.60–0.96). Specifically, participants who met the DGA for protein intakes, whole grains, greens and beans, vegetables, or whole fruits had 20–35% lower odds of having low grip strength than those who did not. Higher compliance to the DGA might reduce the risk for low grip strength as a proxy measure for sarcopenia among U.S. adults, particularly adequate intakes of proteins, whole grains, greens and beans, vegetables, and whole fruits.

## 1. Introduction

Sarcopenia is an accelerated loss of skeletal muscle mass and strength with increasing age which affects >50 million people today and will affect >200 million in the next 40 years worldwide [1]. Sarcopenia can lead to several adverse health outcomes and disabilities, particularly falls, mobility limitations, hip fractures, and increased mortality [2]. Alongside aging, changes in muscle tissues can be caused by a range of chronic diseases including diabetes, arthritis, and stroke [3,4,5], and by unhealthy lifestyles such as smoking, obesity, and physical inactivity [5,6,7]. The 2020 Sarcopenia Definitions and Outcomes Consortium (SDOC) provided an evidence-based, data-driven definition of sarcopenia, which included sex-specific cut-points for muscle weakness and slowness [2]. A decline in muscle weakness can be assessed by grip strength, which is a simple and reliable measure frequently used in research and clinical settings [3].

With increasing age, there are also alterations in dietary intake affecting muscle mass and function [8,9,10,11,12]. Typical changes among older adults are an overall decrease in food intakes, which lead to weight loss resulting in a degree of muscle loss and insufficient intakes of certain nutrients essential in maintaining muscle mass and strength [11,12,13,14,15]. Low protein intake is associated with sarcopenia in several cohort studies, specifically a loss of lean mass [16,17] and reduced grip strength [11,18]. Higher intakes of fruits and vegetables that are rich in antioxidants such as carotenoids, lycopene, lutein, and vitamins are associated with an increased in grip strength, faster gait speed, and with lower risks for sarcopenia among older adults [13,14,19].

Recent studies, however, examine dietary intakes as a mix of foods and nutrients using different types of common eating patterns [20,21,22,23,24,25,26,27,28,29]. For example, adherence to the dietary approaches to stop hypertension (DASH) or to the Nordic diet is positively associated with higher grip strength [20,25]. The Mediterranean diet studied across different populations is related to muscle mass and physical function [20,24,26]. Higher adherence to the Mediterranean diet was positively associated with fat-free mass and leg explosive power, but not with grip strength among United Kingdom women [21], while a positive association with grip strength is observed in active elderly women from Italy and Netherlands [24].

The healthy eating index (HEI) was developed to measure compliance with the Dietary Guidelines for Americans (DGA) and is revised every five years [30,31]. The DGA recommends a healthy diet encouraging eating a variety of vegetables, fruits, whole grains, low-fat dairy, and protein foods within an appropriate calorie level while limiting consumption of saturated fats and trans fats, added sugars, and sodium [30,31]. The recent HEI was updated in 2015 (HEI-2015) and included 13 components assessing foods’ compliance to the DGA 2015–2020 [30,31]. Adherence to HEI is associated with optimal grip strength [23] and with better physical performance of gait speed and knee extension [22]. However, to the best of our knowledge, no studies have examined the updated HEI-2015 and its relationship with grip strength among a US population. Research on the relationship between a recommended eating pattern for Americans and grip strength could inform dietary intervention programs designed for older people and help identify specific food group requirements that may be associated with reduced risk for low grip strength in the US aging population.

The study aims to examine the associations between the HEI-2015 component and its total scores with low grip strength as a proxy measure for sarcopenia in a nationally representative sample of the US adult population. The study hypotheses are (1) participants at the highest level of the total score of HEI-2015 would have lower odds of having low grip strength compared to those at the lowest level, and (2) among the HEI-2015 components, inadequate intakes of fruits, vegetables, and protein foods would be associated with low grip strength.

## 2. Materials and Methods

### 2.1. Study Sample

The study used data from the National Health and Nutrition Examination Survey (NHANES) of 2 cycles: 2011–2014. NHANES is a cross-sectional survey using a stratified multistage probability design. Data on health and nutritional status of non-institutionalized civilians of the U.S. population were collected through a series of interviews, examinations, and laboratory measurements. The National Center for Health Statistics (NCHS) Research Ethics Review Board (ERB) provided the following protocol approval numbers for the presented surveys years: Protocol #2011–17 (NHANES 2011–2012) and Continuation of Protocol #2011–17 (NHANES 2013–2014). Further information on the NHANES database is obtainable at http://www.cdc.gov/nchs/nhanes.htm (accessed on 31 March 2021). The use of de-identified data exempted this study from review by the local Institutional Review Board [32].

From the NHANES 2011–2014 surveys, only adult participants who had available data on grip strength, complete dietary intakes of the two-day dietary interviews, and demographics were included. Participants with a height of >6′5″ or a weight of >450 lb. or pregnant women were not included based on the NHANES protocol. Oversampling, including that of persons aged 60 years and over, was done to improve the reliability and precision of related estimates [33]. Therefore, the final analysis sample comprised of 9006 participants aged 20–80 years with complete available study data.

### 2.2. Study Variables

#### 2.2.1. Dietary Data

We used dietary data of the two 24-h dietary recall interviews. The NHANES used the Automated Multiple Pass Method (AMPM), which is designed to provide an efficient and accurate means of collecting intakes for large-scale national surveys [34]. The AMPM uses a five-step interview: (1) quick list of all foods and beverages consumed the day before the interview (midnight to midnight); (2) list of consumption of foods commonly forgotten; (3) time and eating occasion of each reported food; (4) a detailed description, amount eaten, and additions to the food; (5) additional foods not remembered. Based on the NHANES for reliable dietary data, only participants who met the criteria for inclusion, which was completing the first four steps of the AMPM and all food/beverages consumed were identified [35], were analyzed. The first interview was conducted in-person in the Mobile Examination Center (MEC) by trained interviewers. The second interview was conducted over the telephone three through 10 days later. The response rate in both cycles ranged between 87% and 89%, which provided reliable dietary data for both interviews [36].

##### 2015 Health Eating Index

We used the “Simple HEI Scoring Algorithm-Per two Days” that is implemented in SAS macros and is available from the National Cancer Institute to compute the HEI-2015 score for each participant. The HEI is comprised from thirteen dietary component scores. Maximum scores on the components are either five or 10 points and minimum scores are zero. This index has two sections: adequacy and moderation. Higher scores on the nine adequacy components (total fruits, whole fruits, total vegetables, greens and beans, whole grains, total dairy, total protein foods, seafood and plant proteins, and fatty acid ratio) reflect higher intakes. The other four components are moderation components (refined grains, sodium, added sugars, and saturated fats), which are computed so that higher scores indicate lower intakes [37]. The maximum score for each component is assigned when the person is compliant with the intake level recommended by the 2015–2020 Dietary Guidelines for Americans (DGA) [38]. Each component was scored on a density out of 1000 calories independent of quantity except fatty acids, which used the ratio of unsaturated to saturated fatty acids. Then, the 13 component scores were summed to create a total dietary quality score (range: 0–100) [38]. An HEI total score of >80 is rated as “good”, 51–80 is rated as “needs improvement”, and scores <50 indicate a “poor” diet [38]. However, as in this sample less than 3% were rated as having a good diet (total score > 80), the HEI-2015 total score was divided into quartiles, with quartile_1 having the lowest score and quartile_4 the highest.

For each of the 13 HEI-2015 components, participants were classified as being compliant if they received the maximum component score (e.g., five or 10) and were compared to either noncompliant in case of the following components: refined grains, added sugars, saturated fats, total dairy, fatty acids (e.g., max vs. <max), or to those who received the minimum score of zero in case of greens and beans, total fruits, and seafood and plant proteins (e.g., max vs. zero). Whole grains, sodium, and whole fruits comparisons were made between the minimum score to zero (e.g., min≤ vs. zero). Last, due to insufficient participants at the minimum score of zero in total protein foods and total vegetable, compliant participants were compared to those who were at the lowest 25th percentile.

#### 2.2.2. Low Grip Strength

The grip strength test was taken in 2011–2014 surveys, and the protocol is detailed in the NHANES 2011–2014 Muscle Function Procedures Manual [39]. Briefly, grip strength was measured in kilograms (kg) with the Takei Digital Grip Strength Dynamometer over three trials separated by 60 s and alternating hands. The mean value achieved by all 6 trials was used in the analysis. ‘Low grip strength’ was classified based on the grip strength criteria (<35.5 kg for men; <20 kg for women) defined by the Sarcopenia Definitions and Outcomes Consortium absolute criteria [2].

#### 2.2.3. Covariates

Study covariates included demographics variables such as gender, age in years, race, and ethnicity (Non-Hispanic Whites ((NH)-White reference category), NH-Blacks, Hispanics, Asians, and Other).

Body mass index (BMI) was measured as the weight in kg divided by the square of height in meters(m). As Body Shape Index (ABSI) and grip strength are associated with BMI, we calculated ABSI using the following formula: waist circumference (m) multiplied by height (m)^5/6^ divided by weight (kg)^2/3^ and then adding to the model its logarithmic transformation [40].

Participants reported whether they were taking prescription medications in the past 30 days (yes vs. no) and their medical history which included their physician diagnosis of arthritis, congestive heart failure, coronary heart disease, angina pectoris, heart attack, stroke, cancer, and diabetes (categorized as ‘0’, ‘1’, and ‘2+’ chronic diseases). Health behavioral variables include smoking status (never, former, current smoker) and alcohol use in the past year (never vs. ever).

Physical activity information was based on the Global Physical Activity Questionnaire [41]. Physical activity was divided into two variables: (1) ‘moderate recreational activities’—including activities that cause a small increase in breathing or heart rate such as brisk walking, bicycling, or swimming for at least 10 min continuously (2) vigorous recreational activities—including activities that cause large increases in breathing or heart rate like running or basketball for at least 10 min continuously. Participants who met the World Health Organization 2020 recommendations of at least 150 min of moderate intensity, or at least 75 min of vigorous intensity physical activity, per week, were defined as physically active [42].

### 2.3. Statistical Analysis

All analyses were conducted using Mobile Examination Center (MEC) exam 2-year sampling weights as recommended by the National Center for Health Statistics due to unequal sample selection probability in the NHANES 2011–2014 to account for stratification and clustering due to the complex sample design [43]. For the statistical analyses, Stata 15 [44] was employed. Responses coded as ‘do not know’, ‘refused’, or ‘missing’ in the original NHANES surveys and were treated in these analyses as missing. A type I error level of 0.05 was considered significant throughout the statistical tests.

Descriptive statistics (sample sizes and weighted proportions and means) were used to summarize the characteristics of the study samples. Weighted Chi square test was used to examine the bivariate association between low grip strength and each study covariate separately.

Weighted logistic regression models were used to examine the crude associations between HEI-2015 components and total score with low grip strength separately. For the multivariable analyses, three weighted logistic regression models were built to examine the adjusted associations between HEI-2015 and low grip strength: Model-1 included adjustment for age, gender, race/ethnicity, and education. Model-2 included model 1 covariates plus physical activity, BMI, and comorbidities. Model-3 included model 1 and 2 covariates plus smoking, alcohol, and medication use in the past month.

Last, we examined the interaction term between HEI-2015 total score quartiles and age groups in years (<50, 50–69, 70+) and whether it was significantly associated with low grip strength.

## 3. Results

### 3.1. Subject Characteristics

There were 9006 participants aged 20 years and above in the final study sample of whom 14.4% had low grip strength (Table 1). The prevalence of low grip strength was 15.9% among participants aged 50–69 years and higher in aged 70 years or older (51.0%). On average, participants with low grip strength were significantly older (*p* < 0.001). Males had higher low grip strength prevalence than females (18.4 vs. 10.7%, *p* < 0.001). A smaller proportion of NH-Blacks were classified as having vs. not having low grip strength (8.2 vs. 11.6%, *p* = 0.004), whereas a higher proportion of Asians had low grip strength (9.7 vs. 6.8%, *p* = 0.013). Participants with low grip strength also had lower mean BMI than those without (28.2 vs. 29.2, *p* = 0.005). Lifestyle factors such as smoking, alcohol, and physical activity were significantly associated with low grip strength prevalence. For example, only 9.4% performed vigorous activity among participants with low grip strength compared to 24.4% among those without (*p* < 0.001). Smaller but significant differences were observed among moderate activities (19.2 vs. 23.6%, *p* < 0.001).

Comorbidities were more common among participants with low grip strength (62.6 vs. 32.0%, *p* < 0.001) as well as a higher medication use in the past month in participants with low grip strength (75.6 vs. 56.3%, *p* < 0.001) (Table 1).

### 3.2. HEI-2015 Total and Component Scores by Grip Strength Status

Table 2 shows the weighted proportion of individuals who received the maximum score in each HEI-2015 component and the weighted mean of the total HEI-2015 score by low grip strength status (with vs. without).

Scores for HEI-2015 were total 54.2 ± 13.6 (mean ± SD), and in the lowest and highest quartiles 37.3 ± 5.1 and 72.0 ± 6.5, respectively (data not shown). Mean total HEI-2015 score did not significantly differ between individuals with or without low grip strength (54.7 vs. 54.1, *p* = 0.139) in the overall population; however, among younger adults (aged < 50 years) and in older adults (aged 70+ years) it did, respectively (48.8 vs. 52.1, *p* = 0.005) and (58.3 vs. 59.8, *p* < 0.05).

There were significant differences across low grip strength status in the proportion of participants who were compatible to the DGA 2015–2020 by HEI-2015 component. Overall, 22.5% of the participants were compatible to total DGA 2015–2020 fruits intake, which differed between individuals with or without low grip strength, respectively (25.9 vs. 21.9%, *p* = 0.049). Only 7.4% consumed adequate whole grains per day as recommended by the DGA 2015–2020, which was higher in individuals with low grip strength than those without (10.0 vs. 6.9%, *p* < 0.001). Two-thirds of the participants (64.5%) consumed sufficient protein per day, which did not differ by grip strength status (64.4 vs. 64.6%, *p* = 0.359).

Overall, high prevalence of low grip strength was more pronounced in the age group 70+ years, in particular in participants at the HEI-2015 quartiles 1–3 than those in quartile 4 (53.8 vs. 46.0%, *p* < 0.05) (Figure 1).

### 3.3. HEI-2015 Total and Component Scores and Their Adjusted Associations with Low Grip Strength

Table 3 presents the adjusted odds ratios (OR) between HEI-2015 quartiles and grip strength status (with vs. without) generated from multivariable models (i.e., Models 1–3), which differ in their study covariates. Compared to the lowest HEI-2015 quartile_1, participants in the highest quartile_4 had lower odds of having low grip strength (OR = 0.68, 95%CI: 0.54–0.86) after adjusting for age, gender, race/ethnicity, and education in Model-1. When additional lifestyle covariates were included in the multivariable model such as physical activity, BMI, and comorbidities (Model-2), ABSI, smoking, alcohol, and medication use (Model-3), the final ORs were attenuated but remained significant (Model-2: OR = 0.74; 95% CI: 0.58–0.93; Model-3: OR = 0.76; 95% CI: 0.60–0.96).

Only the highest HEI-2015 quartile was significantly associated with low grip strength (*p* = 0.024), indicating that participants with an average HEI-2015 total score of 72.0 ± 6.5 are less likely to have low grip strength than those with an average score of 37.3 ± 5.1.

When HEI-2015 was examined by its components (Table 3), six components out of 13 were associated with low grip strength: whole fruits, total vegetables, greens and beans, whole grains, and total protein foods and seafood and plant proteins across all the three multivariable models. For example, participants who consumed at least 0.8 oz per 1000 calories a day of seafood and plant proteins reduced their odds of having low grip strength by 35% according to the findings in Model-3. Across all three multivariable models, consuming adequate amount of proteins, whole grains, whole fruits, vegetable, and greens and beans is associated with better grip strength by lowering the odds of having low grip strength by 20–35%.

The additional HEI-2015 components that did not show relationships with sarcopenia in any of the multivariable models were total fruits, dairy, fatty acids, refined grains, sodium, added sugar, and saturated fats. There were no differences in the odds of having low grip strength between participants who consumed at least 1.3 cups per 1000 calories a day of dairy to those who did not. Sufficient amount of total fruits, which includes whole fruits and 100% fruit juice, did not reduce the odds of having low grip strength in this sample.

The crude and adjusted interaction term between quartiles of HEI-2015 total score and age groups (<50, 50–69, 70+) in models 1–3 was not significant (*p* = 0.415–0.807), and therefore the multivariable models 1–3 were not further stratified by age groups.

## 4. Discussion

This cross-sectional study using data from a representative sample of the US adult population showed that high adherence to the 2015–2020 DGA assessed by the HEI-2015, was associated with lower odds of having a low grip strength. After adjusting for demographics, health behaviors, physical activity, and medical conditions, only participants at the highest HEI-2015 quartile_4 were 24% less likely to have low grip strength compared to those at the lowest quartile_1. This study also examined which food categories may benefit muscle strength and found that among the HEI-2015 components, compliances to adequate intake of whole fruits, vegetables, greens and beans, whole grains, protein foods were associated with 20–35% reduction in having low grip strength. Moderate consumption of saturated fats, added sugar, refined grains, and sodium were not associated with low grip strength in the current sample.

Dietary patterns can be determined from the study populations (e.g., healthy diet vs. unhealthy diet) or be a predefined eating pattern (e.g., Mediterranean diet and healthy eating index) [26]. Predefined dietary patterns were previously examined in relation to weak muscle strength and sarcopenia and showed similar results [20,23,27]. The healthy Nordic diet, for instance, is based on healthy, common local Nordic foods, which include high consumption of fruits, vegetables, whole-grain products, fish, and rapeseed oil, and low consumption of red meat and alcohol [20]. A study utilizing data from the Helsinki Birth Cohort Study found that among 1072 participants, women in the highest quartile of the healthy Nordic score had greater grip strength than those in the lowest quartile; however, no such association was observed among men [20]. In the Healthy Aging in Neighborhoods of Diversity across the Life Span (HANDLS) study among 2468 aged 33–71 years, the authors reported positive associations between the higher score in each predefined dietary pattern, DASH and HEI-2010, with better grip strength/BMI ratio [27]. Only one study had specifically examined the HEI-2015 with grip strength conducted among 4010 participants from Iran, aged 35–65 years. The study reported that a higher total score of HEI-2015 is associated with optimal grip strength (OR = 1.26; 95%CI: 1.02–1.62) after adjusting for potential confounders. However, as opposed to our study findings regarding HEI-2015 components, the authors showed that only two components were significant associated with optimal grip strength; intake of whole fruits (OR = 1.10; 95%CI: 1.02–1.18) and added sugar (OR = 1.06; 95%CI: 1.01–1.12) [23].

The Mediterranean diet, another predefined healthy eating pattern, and its relationship with sarcopenia and low grip strength was explored in diverse populations [20,24,26,28,45]. Typically, the Mediterranean diet emphasizes daily intake of non-refined cereals (whole-grain products), fruits, vegetables, legumes, potatoes, fish, and olive oil, moderate alcohol intake, and rare consumption of meat and meat products, poultry, and full-fat dairy products. The benefits of the Mediterranean diet for general health are recognized in the DGA, and therefore many characteristics of the Mediterranean diet are included in the DGA [26,38]. Previous studies showed that higher adherence to the Mediterranean diet is positively associated with better muscle mass and function [21,24,25]. For example, the Mediterranean diet is associated with fat-free mass and leg explosive power, but not with grip strength among United Kingdom older women [21], while among active elderly women in Italy and Netherland, the Mediterranean diet was shown to be positively associated with better grip strength [24]. Among 3675 Korean adults aged 65 or above, the authors reported that higher score in all three eating patterns, including the Korean-HEI, the alternate Mediterranean diet, and the DASH, were associated with 32–53% reduction in the risk of having low grip strength (i.e., corresponded to the lowest 20th percentile of the study population) [25].

Other dietary studies examining the relationships between diet and grip strength focused on intake of certain food categories (e.g., fruits and vegetables) [13,14,19,46,47]. In the current study, the adjusted model showed that those who consumed more vegetables were less likely to have low grip strength. This finding is consistent with other studies examining vegetables as part of a diet. In a randomized controlled intervention study, it was demonstrated that five or more portions/day of fruits and vegetables for 16 weeks improved grip strength among older adults [14]. In a sample of 84 healthy elderly women with active lifestyles (60 years of age or older), those who consumed at least two servings sizes of vegetables (equal to 2 cups) had 20% higher grip strength (>20 kg) after adjusting for BMI [24]. A cohort of 432 middle-aged African Americans followed for six years, found that vegetable intake other than carrots, salads, or potatoes was associated with better grip strength, while fruit juice was associated with worse grip strength after adjusting for physical activity [47]. In our current findings, while higher intake of whole fruit was associated with lower risk of having low grip strength, intake of total fruits, including whole fruits and 100% fruit juice, was not, suggesting that consumption of 100% fruit juice alone might be associated differently with low grip strength than whole fruits or may not be advantageous in reducing the risk for low grip strength.

Vegetables and fruits are high sources of antioxidants (e.g., carotenoids, vitamin C, vitamin E, selenium, magnesium, and other polyphenols). Higher intakes of total antioxidants such as carotenoids, lycopene, and lutein + zeaxanthin were associated with an increased in grip strength and with faster gait speed among participants aged 33–88 years who were followed for 12 years at the Framingham Offspring cohort after adjustment for age, sex, height, BMI, physical activity, energy intake (residual method), current smoking, and multivitamin supplement use [13]. Among 68,002 participants (age 63.8 ± 2.7 years) from UK Biobank after adjustment for age, sex, month of assessment, ethnicity, deprivation index, height, comorbidities, and total energy intake, positive associations between higher retinol and magnesium intake and better grip strength in both sexes were observed [46]. Furthermore, among women, positive associations were also observed between higher intake of high sources of antioxidants, such as vitamins E, C, B12, and folate, and increased in handgrip strength [46].

The DGA aimed to primarily promote health by meeting daily nutrient needs and reducing the risk of certain chronic diseases associated with chronic inflammation (e.g., cardiovascular disease and cancer) [48]. Chronic inflammation and oxidative stress are principal mechanisms also involved in the pathogenesis of sarcopenia in aging adults by triggering catabolism and increasing protein turnover in skeletal muscle, which leads to reduction in the overall muscle strength [13,49,50]. Therefore, the association between high HEI total score and low grip strength via high consumption of fruits and vegetables rich in antioxidants may explain the DGA myoprotective effects on muscle strength decline [13,26,49,50].

Protein, a key nutrient for muscle tissues, has been extensively examined in relation to sarcopenia and low grip strength [10,11,12,16,17,18]. The current study deduced that sufficient protein intake, particularly seafood and plant proteins as part of an overall healthy diet, might prevent muscle strength loss in adults. In the Framingham Offspring cohort, among participants aged 29–85 years, the authors found that those in the lowest quartiles of protein intake lost grip strength by 0.17% to 0.27% per year, while those in the highest quartiles of protein intake improved their grip strength by 0.52% to 0.60% per year, suggesting that increasing intake of protein may help maintain muscle strength in adults and older adults [11]. In general, dietary proteins are key sources of essential amino acids needed to synthesize muscle protein across all age groups. Therefore, insufficient protein intake in the diet as well as low physical activity can lead to catabolism and loss of muscle tissues, which is a more prevalent phenomenon among older adults [11,12,13,14,15].

Dairy foods, another primary source for proteins and essential amino acids, examined as an individual HEI component, did not show a significant association with low grip strength in the current study. Our finding was not consistent with another study conducted among 1456 women, aged 70–85 in Australia [51]. Compared to those in the first tertial with the lowest dairy intake, women in the highest tertial had significantly greater grip strength [51]. The study findings suggest that different sources of protein may affect muscle strength differently among adults; therefore, more studies are needed to examine specific sources of protein and amino acids and their relationships with muscle strength and sarcopenia.

The current study has limitations worth mentioning. First, the NHANES cross-sectional data only confirm associations between HEI-2015 and low grip strength and not causality. The dietary data were based on a 24-h recall, which may have resulted in measurement errors. However, the analysis included two nonconsecutive 24-h recalls used by NHANES, which remains a valid tool for assessing usual diet and describing intake among the US adult populations [25,34]. This study has also some strengths. The study utilized data from the NHANES, which employed standardized protocols and rigorous quality control in data collection and reporting among all participants. Another key strength was the use of data from a broad representative sample of the US adult population, thus providing some degree of generalizability. Moreover, this sample is made up of young, middle-aged, and older adults, adding to the literature on sarcopenia and low grip strength, which primarily conducted in adults aged 60 years or older [2,12,14]. Furthermore, the NHANES is a comprehensive dataset that allowed adjustment for demographics, health behaviors, physical activity, and medical conditions such as diabetes, arthritis, and cardiovascular diseases, which are important covariates related to both dietary intakes and low grip strength [4,5,6,7].

## 5. Conclusions

The findings suggest that higher compliance to the 2015–2020 DGA might reduce the risk for low grip strength among US adults, particularly adequate intakes of protein foods, greens and beans, whole grains, vegetables, and whole fruits. Forthcoming prospective studies or interventions are needed to confirm such findings. Once the relationship between HEI-2015 and grip strength is confirmed, this could help in the development of tailored intervention program to prevent low grip strength and sarcopenia in the US aging population.

## Figures and Tables

**Figure 1 nutrients-13-03358-f001:**
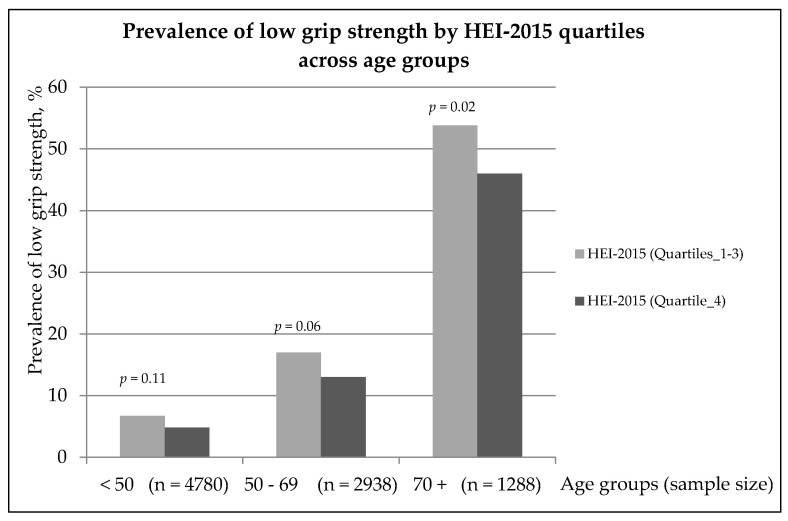
Low grip strength prevalence by HEI-2015 quartiles across age groups.

**Table 1 nutrients-13-03358-t001:** Study characteristics by grip strength status in U.S. adults (Age ≥ 20 years) NHANES 2011–2014, *n* = 9006.

	Total *n* (%) *n* = 9006	Low Grip Strength Status *n* (%)	
		With *n* = 1693 (14.4%)	Without *n* = 7313 (85.6%)
**Gender**			
Male	4461(48.9)	1108 (62.0)	3384 (46.6) ^a^
Female	4545 (51.1)	622 (38.0)	3959 (53.4)
Age^+^ (years)	47.0 (16.8)	60.0 (20.8)	44.9 (15.1) ^a^
Age groups			
<50 years	4780 (55.3)	369 (24.8)	4411 (60.7) ^a^
50–69 years	2938 (33.3)	616 (36.3)	2322 (32.8)
70+ years	1288 (11.4)	708 (39.7)	580 (6.5)
Race/Ethnicity			
NH-Whites	3787 (68.0)	739 (67.8)	3048 (68.1) ^b^
Hispanics	1830 (13.6)	387 (14.3)	1443 (13.5)
NH-Blacks	2097 (11.1)	280 (8.2)	1817 (11.6)
Asians/Others	1292 (7.3)	287 (9.7)	1005 (6.8)
Education			
<High School	1855 (14.5)	517 (22.5)	1338 (13.2) ^a^
≥High School	7151 (85.5)	1176 (77.5)	5975 (86.8)
Physical Activity			
Moderate recreational activities ≥ 150 min/week	1888 (22.9)	301 (19.2)	1587 (23.6) ^a^
Vigorous recreational activities ≥ 75 min/week	1765 (22.1)	143 (9.4)	1622 (24.4) ^a^
BMI (kg/m^2^)	29.0 (6.9)	29.2 (6.8)	28.2 (7.3) ^b^
ABSI (log)	−2.51 (0.06)	−2.51 (0.05)	−2.47 (0.07) ^a^
Alcohol Use			
Never	1699 (15.4)	438 (23.7)	1261 (14.0) ^a^
Ever	7307 (84.6)	1255 (76.3)	6052 (86.0)
Smoking Status			
Never	5090 (56.3)	893 (54.2)	4197 (56.7) ^a^
Former	2073 (24.1)	522 (30.0)	1551 (23.1)
Current	1839 (19.6)	277 (15.8)	1562 (20.2)
Comorbidities			
None	5600 (63.6)	647 (37.8)	4953 (68.0) ^a^
1	2073 (23.0)	502 (29.9)	1571 (21.9)
2 or more	1333 (13.4)	544 (32.3)	789 (10.1)
Medication Use in the past month	5091 (59.1)	1247 (75.6)	3844 (56.3) ^a^

ABSI, Body Shape Index; BMI, Body Mass Index; NH, Non-Hispanics; NHANES, National Health and Nutrition Examination Survey; U.S, United States. ^a^
*p*-value < 0.001, ^b^
*p*-value < 0.01. + Weighted Mean (SD).

**Table 2 nutrients-13-03358-t002:** Weighted proportions of individuals who received maximum score in each of the Health Eating Index-2015 (HEI-2015) component, and weighted mean of HEI-2015 total scores, by low grip strength status. U.S. adults (Age 20–80 years) NHANES 2011–2014. *n* = 9006.

Components	HEI–2015 ^1^				Low Grip Strength Status% (95% CI)		
	Max Points	Standard for Maximum Score	Standard for Minimum Score-0	Total *n* = 9006	With *n* = 1693	Without *n* = 7313	*p*-Value
**Adequacy:**							
Total Fruits ^2^	5	≥0.8 cup equiv.per 1000 kcal	No Fruits	22.5 (20.1–24.2)	25.9 (23.0–28.9)	21.9 (20.2–23.8)	0.049
Whole Fruits ^3^	5	≥0.4 cup equiv. per 1000 kcal	No Whole Fruits	33.4 (31.5–35.3)	36.5 (33.5–39.6)	32.9 (30.8–35.0)	0.142
Total Vegetables ^4^	5	≥1.1 cup equiv. per 1000 kcal	No Vegetables	24.5 (23.0–26.0)	26.4 (23.5–29.5)	24.2 (22.5–25.9	0.129
Greens and Beans ^4^	5	≥0.2 cup equiv. per 1000 kcal	No Greens & Beans	24.9 (23.6–26.3)	22.1 (19.4–25.1)	25.4 (23.9–26.9)	0.050
Whole Grains	10	≥1.5 oz equiv. per 1000 kcal	No Whole Grains	7.4 (6.8–8.0)	10.0 (8.5–11.7)	6.9 (6.3–7.6)	0.001
Total Dairy ^5^	10	≥1.3 cup equiv. per 1000 kcal	No Dairy	13.4 (12.3–14.6)	13.3 (11.1–15.9)	13.4 (12.3–14.6)	0.912
Total Protein Foods ^4^	5	≥2.5 oz equiv. per 1000 kcal	No Protein Foods	64.5 (62.8–66.2)	64.4 (62.1–66.6)	64.6 (62.5–66.6)	0.359
Seafood and Plant Proteins ^6^	5	≥0.8 oz equiv. per 1000 kcal	No Seafood or Plant Proteins	40.5 (38.5–42.4)	37.6 (34.6–40.8)	40.9 (39.0–42.9)	0.005
Fatty Acids ^7^	10	(PUFAs + MUFAs)/SFAs ≥2.5	≤1.2	16.8 (15.6–18.1)	18.4 (16.2–20.8)	16.6 (15.3–17.9)	0.125
**Moderation:**							
Refined Grains	10	≤1.8 oz equiv. per 1000 kcal	≥4.3 oz equiv. per 1000 kcal	24.4 (23.3–25.6)	25.6 (23.3–27.9)	24.2 (22.9–25.5)	0.290
Sodium	10	≤1.1 g. per 1000 kcal	≥2.0 g. per 1000 kcal	5.4 (4.7–6.1)	5.8 (4.6–7.3)	5.3 (4.7–6.0)	0.453
Added Sugars	10	≤6.5% of energy	≥26% of energy	23.7 (22.4–25.1)	24.6 (21.9–27.3)	23.6 (22.2–25.1)	0.524
Saturated Fats	10	≤8% of energy	≥16% of energy	17.7 (16.4–19.1)	18.7 (16.0–21.7)	17.5 (160–19.1)	0.452
**Total Score**	100			**54.2 (13.6)**	**54.7 (15.0)**	**54.1 (13.3)**	**0.139**
Total Score by age group		<50 years		51.8(13.0)	48.8 (14.0)	52.1 (13.0)	0.005
		50–69 years		56.3 (13.3)	54.7 (14.5)	56.6 (13.0)	0.061
		70≤ years		59.0 (14.3)	58.3 (14.6)	59.8 (13.8)	0.048

^1^ Intakes between the minimum and maximum standards are scored proportionately. ^2^ Includes 100% fruit juice. ^3^ Includes all forms except juice. ^4^ Includes legumes (beans and peas). ^5^ Includes all milk products, such as fluid milk, yogurt, and cheese, and fortified soy beverages. ^6^ Includes seafood, nuts, seeds, soy products (other than beverages), and legumes (beans and peas). ^7^ Ratio of poly- and monounsaturated fatty acids (PUFAs and MUFAs) to saturated fatty acids (SFAs). HEI-2015, Healthy Eating Index 2015; NHANES, National Health and Nutrition Examination Survey; U.S, United States.

**Table 3 nutrients-13-03358-t003:** Odds Ratios between health eating index 2015 (HEI-2015) components and total scores and low grip strength. U.S. adults (Age 20–80 years) NHANES 2011–2014. *n* = 9006.

Healthy Eating Index-2015 Components	Model-1 OR (95% CI) ^+^	Model-2 OR (95% CI) ^++^	Model-3 OR (95% CI) ^+++^
**Adequacy**			
Total Fruits ^2^**	0.83 (0.63–1.10)	0.92 (0.71–1.19)	0.89 (0.67–1.18)
Whole Fruits ^3^***	0.77 (0.63–0.94)	0.81 (0.67–0.98)	0.79 (0.63–0.98)
Total Vegetables ^4^ ****	0.76 (0.64–0.90)	0.78 (0.65–0.94)	0.80 (0.67–0.95)
Greens and Beans ^4^**	0.74 (0.60–0.91)	0.78 (0.63–0.97)	0.78 (0.62–0.98)
Whole Grains ***	0.77 (0.67–0.89)	0.78 (0.67–0.91)	0.79 (0.66–0.94)
Total Dairy ^5^*	1.02 (0.82–1.27)	1.00 (0.80–1.26)	0.99 (0.79–1.23)
Total Protein Foods ^4^****	0.74 (0.61–0.91)	0.76 (0.62–0.93)	0.78 (0.64–0.96)
Seafood and Plant Proteins ^6^**	0.59 (0.48–0.72)	0.61 (0.50–0.75)	0.65 (0.53–0.80)
Fatty Acids ^7^*	1.05 (0.90–1.23)	1.07 (0.92–1.24)	1.12 (0.95–1.30)
**Moderation**			
Refined Grains *	0.95 (0.84–1.06)	0.95 (0.84–1.07)	0.99 (0.88–1.14)
Sodium ***	1.06 (0.84–1.19)	0.96 (0.80–1.15)	0.98 (0.80–1.19)
Added Sugars *	0.87 (0.73–1.03)	0.90 (0.74–1.09)	0.91 (0.74–1.11)
Saturated Fats *	1.01 (0.80–1.27)	1.06 (0.85–1.33)	1.07 (0.84–1.36)
**Total Score** (HEI-2015 _Q4vs.Q1_)	0.68 (0.54–0.86)	0.74 (0.58–0.93)	0.76 (0.60–0.96)

^1^ Intakes between the minimum and maximum standards are scored proportionately. ^2^ Includes 100% fruit juice. ^3^ Includes all forms except juice. ^4^ Includes legumes (beans and peas). ^5^ Includes all milk products, such as fluid milk, yogurt, and cheese, and fortified soy beverages. ^6^ Includes seafood, nuts, seeds, soy products (other than beverages), and legumes (beans and peas). ^7^ Ratio of poly-and monounsaturated fatty acids (PUFAs and MUFAs) to saturated fatty acids (SFAs). ^+^ Adjusted for age, gender, race, and ethnicity and education. ^++^ Adjusted for age, gender, race and ethnicity, education, moderate and vigorous physical activities, Body mass index (BMI), and comorbidities ^+++^
*n* = 8603 Adjusted for age, gender, race and ethnicity, education, moderate and vigorous physical activities, BMI, comorbidities, Body Shape Index (ABSI-log) smoking status, alcohol use, and medication use in the past month. HEI-2015, Healthy Eating Index 2015; NHANES, National Health and Nutrition Examination Survey; U.S, United States. * Comparisons between participants receiving the maximum score of either five or 10 to those receiving below it (e.g., max vs. <max). ** Comparisons between participants receiving the maximum score of either five or 10 to those receiving the score of zero (e.g., max vs. zero). *** Comparisons between participants receiving the above the minimum score to those receiving the score of zero (e.g., min < vs. zero). **** Comparisons between above and below the 25th percentile.

## Data Availability

The NHANES database is available at http://www.cdc.gov/nchs/nhanes.htm (accessed on March 2021).

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
