# Peer review of "Healthy Eating Index-2015 Is Associated with Grip Strength among the US Adult Population"

_nutrients, 2021, doi:10.3390/nu13103358_

Round 1

Reviewer 1 Report

  1. This is an analysis of total HEI-2015 score with grip strength. Strength is an important indicator of sarcopenia as referenced in the Bhasin paper (both weakness and slowness should be included in the definition of sarcopenia).  Unless there is a consensus definition aligned with using grip strength as the sole indicator of sarcopenia, “Sarcopenia” should be removed from the title and throughout the paper and replace with “Grip strength.”  Also see Cruz-Jentoft 2019 which is listed as a reference but not cited within the manuscript ("probable sarcopenia").
  2. These data indicate participants with lower grip strength were significantly older. This justifies analysis within subgroups – older adults.  For example, 51+ years, and/or 71+ years, if the discussion is sarcopenia.  I agree with the comment on line 227 that older adults may differ in their eating habits, so this is another reasons to include analysis within older subgroups.
  3. Healthy eating index only provides subcategories of foods (three of which provide significant amounts of protein). Conclusions about any specific foods cannot be made, unless additional analysis was conducted.  For example, line 295 states “This study also examined which specific food components may benefit muscle strength,” but only food categories are identified by HEI-2015.

Abstract:

  1. 14.4% of those aged 20-80 years had sarcopenia. Please provide data for older adults.  See comment above – insight for older age groups needs to be provided, even if the number is small.  Indeed, the introduction focuses on older adults.

Introduction:

  1. Line 66: Did reference 23 define sarcopenia with grip strength? Or just report HEI-15 association with grip strength?  Recommend being precise with using the term “sarcopenia” throughout the manuscript as this term has been evolving for some time.
  2. Line 69: loss of muscle strength not measured. These are cross-sectional data.

Results:

  1. Table 2:
    1. Typos in headline (Healthy Eating Index-2015)
    2. This table should also include a subgroup analysis of older adults
    3. Is the adequacy per 1000 kcal?
  2. Figure 1: Numbers can be reported without this figure.
  3. Line 264: Typo? As written, it appears that higher healthy eating index is associated with “sarcopenia.”
  4. Line 271: Same comment - confusing as written.

Discussion:

  1. Delete lines 286-289
  2. Line 346: Sarcopenia was not self-reported.
  3. Lines 389-391: Please clarify – the wording is confusing here. 2/3 of participants consumed enough protein from these foods named?  What was the percentage from these foods? (The percentage of protein from dairy foods follows in this discussion).  Could this be divided further?  The foods named include foods from the total protein foods category, as well as the seafood and plant category.  Was further analysis at the food level conducted?  If so, please include in methods section.
  4. Lines 393-394: Recommend also including mention of low physical activity.

Author Response

Thank you for reviewing our manuscript and providing your valuable comments

Comments and Suggestions for Authors

  1. This is an analysis of total HEI-2015 score with grip strength. Strength is an important indicator of sarcopenia as referenced in the Bhasin paper (both weakness and slowness should be included in the definition of sarcopenia).  Unless there is a consensus definition aligned with using grip strength as the sole indicator of sarcopenia, “Sarcopenia” should be removed from the title and throughout the paper and replace with “Grip strength.”  Also see Cruz-Jentoft 2019 which is listed as a reference but not cited within the manuscript ("probable sarcopenia").
    Response:
    We replaced ‘sarcopenia’ with ‘low grip strength’ or ‘grip strength’ in the pertinent places in the text.  The citation Cruz-Jentoft 2019 was removed from the reference list. Thank you for catching this typo.
  2. These data indicate participants with lower grip strength were significantly older. This justifies analysis within subgroups – older adults.  For example, 51+ years, and/or 71+ years, if the discussion is sarcopenia.  I agree with the comment on line 227 that older adults may differ in their eating habits, so this is another reasons to include analysis within older subgroups.
    Response: We included additional analyses within age subgroups of <50, 50-69, and 70+. Please see the results section (Tables 1 and 2) and in the text highlighted in yellow.
  3. Healthy eating index only provides subcategories of foods (three of which provide significant amounts of protein). Conclusions about any specific foods cannot be made, unless additional analysis was conducted.  For example, line 295 states “This study also examined which specific food components may benefit muscle strength,” but only food categories are identified by HEI-2015. 
    Response: We acknowledge this comment and correct the sentence accordingly. 

Abstract:

  1. 14.4% of those aged 20-80 years had sarcopenia. Please provide data for older adults.  See comment above – insight for older age groups needs to be provided, even if the number is small.  Indeed, the introduction focuses on older adults.
    Response: Prevalence of low grip strength for older adults was added to the results section – in the first paragraph highlighted in yellow page 5 (lines 206-209) and page 8 (228-239) .

Introduction:

  1. Line 66: Did reference 23 define sarcopenia with grip strength? Or just report HEI-15 association with grip strength?  Recommend being precise with using the term “sarcopenia” throughout the manuscript as this term has been evolving for some time.
    Response:
    This sentence was revised and the term sarcopenia was removed lines 70-72.

  2. Line 69: loss of muscle strength not measured. These are cross-sectional data.
    Response:
    This sentence was corrected accordingly.  Lines 73

Results:

  1. Table 2:
    1. Typos in headline (Healthy Eating Index-2015)
      Response: Thank you we corrected it.
    2. This table should also include a subgroup analysis of older adults
      Response: We included a subgroup analysis by age groups for adults, older adults and elderly (<50, 50-69, 70+) pages 6 -7 Tables 1 and  2. We also provided a new figure on the differences in low grip strength (%) by HEI quartiles across age groups.
      Page 8.
    3. Is the adequacy per 1000 kcal?
      Response:
      Yes. This was corrected in the text and table. Thank you. Line 298

  2. Figure 1: Numbers can be reported without this figure.
    Response:
    The figure was removed and only the numbers are reported now. Line 258-260

  3. Line 264: Typo? As written, it appears that higher healthy eating index is associated with “sarcopenia.” Line 271: Same comment - confusing as written.
    Response:
    These lines were corrected. High score at the HEI was associated with reducing the odds (risk) of having low grip strength. Since it is a cross sectional and we calculates effect estimates with Odds Ratios (ORs) the terms odds is being used in the interpretation.  Although using the term ‘risk’ might be more intuitive but since we did not calculate risk ratios (RR) we did not use it in the interpretation.

Discussion:

  1. Delete lines 286-289
    Response:  Done. Thank you
  2. Line 346: Sarcopenia was not self-reported.
    Response: Did you mean that the dietary data were not self reported?. We revised this sentence.
  3. Lines 389-391: Please clarify – the wording is confusing here. 2/3 of participants consumed enough protein from these foods named?  What was the percentage from these foods? (The percentage of protein from dairy foods follows in this discussion).  Could this be divided further?  The foods named include foods from the total protein foods category, as well as the seafood and plant category.  Was further analysis at the food level conducted?  If so, please include in methods section. 
    Response: We have removed this part in the text because further dividing protein sources beyond what is done in the HEI index was not conducted in this study and we only used the original components of HEI-2015.
  4. Lines 393-394: Recommend also including mention of low physical activity.
    Response: Thank you and we did mention low physical activity.

Reviewer 2 Report

B"sd

Author Response

Thank you for reviewing our paper and providing your suggestions.

We read your paper1 with great interest and learned about the significant relationship between grip strength and ABSI (allometric index of WC corrected for BMI) and mortality. Following your suggestion, we added ABSI to the models by taking its logarithm. Some of our findings were altered and therefore we divided the HEI-2015 total score instead of tertiles into quartiles.

References:

1. Krakauer NY, Krakauer JC. Association of Body Shape Index (ABSI) with Hand Grip Strength. Int J Environ Res Public Health. 2020 Sep 17;17(18):E6797. doi: 10.3390/ijerph17186797. PMID: 32957738
